# Selection for Improved Water Efficiency in Broiler Breeder Lines Does Not Negatively Impact Immune Response Capabilities to Gram^−^ and Gram^+^ Bacterial Components and a Killed-*Salmonella* Enteritidis Vaccine

**DOI:** 10.3390/vetsci12030279

**Published:** 2025-03-16

**Authors:** Jossie M. Santamaria, Chrysta N. Beck, Sara K. Orlowski, Maricela Maqueda, Walter G. Bottje, Gisela F. Erf

**Affiliations:** Department of Poultry Science, Center of Excellence for Poultry Science, University of Arkansas System Division of Agriculture, Fayetteville, AR 72701, USA; cnbeck@uark.edu (C.N.B.); orlowski@uark.edu (S.K.O.); mamaqued@uark.edu (M.M.); wbottje@uark.edu (W.G.B.)

**Keywords:** broiler breeders, water efficiency, lipopolysaccharide, peptidoglycan, *Salmonella* vaccine, inflammatory responses, inflammatory cytokines, ROS, AGP-1, antibody responses

## Abstract

Water scarcity remains a critical challenge globally. Addressing this pressing issue is important to ensure food supply and water sustainability for the growing population and future generations. Water-efficient broiler breeder chickens are being developed at the University of Arkansas to promote efficient and sustainable water usage in poultry farming. For this, broiler breeders from a modern random-bred (MRB) control population were divergently selected based on low or high water conversion ratio (LWCR or HWCR, respectively), where WCR indicates water consumption per pound of body weight gain. This study assessed the impact of selection for water efficiency on the local (tissue/cellular) and systemic (blood) immune responses to components of Gram^−^ and Gram^+^ bacteria and whole bacteria, namely, lipopolysaccharide (LPS), peptidoglycan (PGN), and a killed-*Salmonella* Enteritidis vaccine (SEV). Our results indicate that selection for improved water efficiency in the LWCR line did not negatively affect local and systemic cellular and humoral immune response capabilities to LPS and PGN bacterial components and killed SEV compared to the MRB control population. Overall, the robust and balanced immune system capabilities observed in the LWCR line with improved water efficiency show that, although they underwent selection pressure, their innate and adaptive defenses were not compromised under improved water sustainability practices.

## 1. Introduction

Modern agriculture is among the main human activities that require large volumes of fresh water. Food production alone requires up to 70% of available fresh water, and in developing countries, demand can reach up to 80 to 90% [1,2,3]. However, water scarcity poses a challenge to food production systems, specifically because of the unequal rainfall distribution, which the rapid changes in climate conditions can exacerbate [4]. It is projected that there will be over an 80% increase in water scarcity worldwide for 19 crops in the coming decades [5]. The anticipated population growth will result in an additional demand for fresh water, requiring considerably higher amounts of this vital nutrient for food production [6]. Therefore, water sustainability practices must be fully incorporated into all agricultural systems and become more efficient to ensure food security for the growing population and future generations.

In the last decades, poultry meat and eggs have become a food staple due to their high nutritional value and accessibility worldwide [7]. Additionally, chicken meat and egg consumption are estimated to increase in developing countries where water scarcity remains a significant problem [8]. Establishing sustainable and efficient poultry farming programs will be necessary to provide high-quality poultry products to the 9 billion individuals expected to be present by 2050 [9]. However, in efforts to contribute to the sustainable use of water resources in poultry farming, the development of water-efficient broiler breeder chickens was initiated, and lines are currently maintained at the University of Arkansas [10,11]. The divergent selection for water efficiency in broiler breeders was started from a modern random-bred (MRB) control population by using a water conversion ratio (WCR) trait distinction where the weight of consumed water is divided by body weight gain (BWG); hence, breeders with a low (L)WCR consume less water (higher water efficiency), while those with high (H)WCR consume more water per pound of BWG (lower water efficiency).

The health and well-being of broiler breeders and their offspring depend on proper immune system functions, especially since they have undergone selection pressure to manifest other desired production traits. However, there is a lack of information on how genetic selection for WCR impacts local and systemic and innate and adaptive immune functions in broiler breeders. A poorly balanced immune system can increase susceptibility to infectious diseases, affecting growth, performance, and well-being [12,13,14]. Therefore, it is important to gain insights into the impact of trait selection on various aspects of the immune system to ensure that while water efficiency in broiler breeders is improved, the birds’ ability to mount effective immune responses is not compromised.

This study implemented a dual-window approach to monitor and assess the immune capabilities of the MRB, LWCR, and HWCR broiler breeder lines. The dual-window approach is comprised of intradermal (i.d.) injections of multiple growing feather (GF)-pulps—a skin derivative and complex tissue—with test materials. Concurrent sampling of injected GF and peripheral blood before and at various times after i.d. GF-pulp injections enables the assessment of local (tissue/cellular) and systemic (blood) immune responses to test materials in chickens [15]. Previously, different longitudinal studies that used the dual-window approach provided valuable insights into the innate and adaptive immune responses to GF-pulp injections of diverse microbial-associated molecular patterns (MAMPs) as well as inactivated vaccines in meat- and egg-type chickens [16,17,18,19].

To gain insight into the effects of selection for water efficiency on innate and adaptive immune response capabilities in the MRB, LWCR, and HWCR broiler breeder lines, the local tissue/cellular and systemic immune responses to MAMPs from Gram-negative and Gram-positive bacteria [i.e., lipopolysaccharide (LPS) from *Salmonella* Typhimurium and peptidoglycan (PGN) from *Staphylococcus aureus*, respectively], as well as to a first immunization with a formalin-killed *Salmonella enterica* serovar Enteritidis vaccine (SEV), were examined. Specifically, three longitudinal studies were conducted in 10- to 14-week-old MRB, LWCR, and HWCR broiler breeders. Laboratory measurements included the assessment of GF-pulp and peripheral blood leukocyte population profiles, GF-pulp relative cytokine/chemokine mRNA expression and ROS generation, as well as plasma concentrations of α1-acid glycoprotein (AGP-1), and relative levels of SE-specific IgM, IgY(G), and IgA in the plasma.

## 2. Materials and Methods

### 2.1. Ethical Statement and Experimental Animals

The protocols and procedures involving animals in this study were approved by the University of Arkansas System Division of Agriculture Animal Care and Use Committee (Ag-IACUC protocol 23019). This study consisted of 3 experimental trials involving juvenile broiler breeder birds from 3 distinct broiler lines. The lines utilized include selected broiler lines in their 4th generation of divergent selection for water conversion ratio (low water conversion ratio—LWCR and high water conversion ratio—HWCR) [11] and their modern random-bred (MRB) [10] control population from which the lines were initially selected. These research lines were selected and maintained at the University of Arkansas Poultry Research farm. Technology used in the selection of the LWCR and HWCR lines are outlined in [11]. Briefly, selection for WCR occurs in a 2-phase system for each generation. Day-old chicks from 11 sire families per line are placed in floor pens and WCR is recorded for 6 weeks for selection purposes on ad libitum fed birds. For this study, following the initial selection of the lines from hatch to 6 weeks, all birds were placed on a broiler breeder controlled feeding program for 16 weeks. Each bird was allocated a set amount of a pullet grower pelleted feed per day that was formulated to meet or exceed NRC requirements. Feed allocations were adjusted by age and average body weight of the birds. Birds were kept on a photo-schedule of 8 h of light and 16 h of dark, and water was provided ad libitum. Pens (10 ft × 10 ft) were equipped with a nipple water line, 2 trough feeders, and fresh pine shavings at placement. Broiler breeders were reared sex-separate; however, lines were mixed within pens. For all trials, treatment application and sample collection were started and completed prior to daily feeding.

### 2.2. Experimental Design (Individual Methods Are Outlined in Section 2.3, Section 2.4, Section 2.5, Section 2.6, Section 2.7, Section 2.8 and Section 2.9)

#### 2.2.1. Trial 1—Local and Systemic Inflammatory Responses to Intradermal (i.d.) GF-Pulp Injections of Lipopolysaccharide (LPS) in 10-Week-Old Male Broiler Breeders from the MRB, LWCR, and HWCR Lines

The pulps of growing feathers (GFs) of 10-week-old broiler breeder males from the MRB, LWCR, and HWCR lines received i.d. injections of either 10 μL PBS (vehicle; 3 birds/line) or 10 μL of LPS from *Salmonella* Typhimurium (Sigma, St. Louis, MO, USA) prepared in PBS at 100 μg/mL (1 μg LPS/GF; 12 GF/bird; 6 birds/line). Four GFs were collected from each bird before injection (0 h) and at 6 and 24 h post LPS injection. At each timepoint, 2 GFs were flash-frozen for later RT-qPCR analysis, and the other 2 were placed in ice-cold PBS to prepare single-cell suspensions for ROS generation assay and to assess leukocyte population profiles by fluorescence-based flow cytometry (FACS). Additionally, heparinized blood samples were collected before (0 h) and at 6 and 24 h for whole blood leukocyte population analysis by FACS, and to determine the concentration of plasma alpha-1-acid glycoprotein (AGP-1) by ELISA.

#### 2.2.2. Trial 2—Local and Systemic Inflammatory Responses to i.d. GF-Pulp Injections of Peptidoglycan (PGN) in 11-Week-Old Male Broiler Breeders from the MRB, LWCR, and HWCR Lines

The pulps of GFs of 11-week-old broiler breeder males from the MRB, LWCR, and HWCR lines received i.d. injections of either 10 μL PBS (vehicle; 3 birds/line) or 10 μL of PGN from *Staphylococcus aureus* (Invivogen, San Diego, CA, USA) prepared in PBS at 100 μg/mL (1 μg PGN/GF; 16 GF/bird; 6 birds/line). Four GFs were collected from each bird before (0 h) and at 0, 6, 24, 48, and 72 h post-PGN injection. At each timepoint, 2 GFs were flash-frozen for later RT-qPCR analysis, and the other 2 were placed in ice-cold PBS to prepare single-cell suspensions to assess leukocyte population profiles by FACS. Additionally, heparinized blood samples were collected before (0 h) and at 6 and 24 h for whole blood leukocyte population analysis by FACS, and to determine the concentration of plasma AGP-1 by ELISA.

#### 2.2.3. Trial 3—Local and Systemic Innate and Adaptive Immune Responses to a First i.d. GF-Pulp Immunization with a Formalin-Killed *Salmonella* Enteritidis Vaccine (SEV) in 14-Week-Old Broiler Breeder Pullets from the MRB, LWCR, and HWCR Lines

Thirty-six 14-week-old feed-restricted broiler breeder pullets from the HWCR, LWCR, and MRB lines (*n* = 6 birds/line/treatment) received a primary vaccination by i.d. GF-pulp injections of 10 μL PBS (vehicle) or a formalin-killed *Salmonella* Enteritidis vaccine (SEV; 10^8^ CFU/mL) prepared in PBS (20 GFs per bird of each treatment). FACS was used to assess the leukocyte population profiles in the GF-pulp and in heparinized blood before (0 h) and at 6, 24, 48, and 72 h post-GF-pulp injection. Additional heparinized blood samples were collected before (0 d) and 3, 5, 7, 10, 14, 21, and 28 d after GF-pulp injections to measure relative plasma levels of SEV-specific IgM, IgG, and IgA antibodies.

### 2.3. Growing Feather Bioassay: Intradermal Injections and Sample Collection

The growing feather (GF) bioassay was conducted as previously described [16]. The availability of same-size GFs for i.d. pulp injections was secured by gently plucking GFs from each breast tract of the broiler breeders and allowing the GFs to regenerate for 18 days. Following collection of pre-injection samples (0 h), GFs were prepared for i.d. pulp injection by cutting off the top 2–3 mm (above the pulp epidermis) of the GFs’ sheath with scissors. Injections were conducted using 0.03 mL graduated syringes with 31-gauge × 8 mm needles (BD, Franklin Lakes, NJ, USA). The needle was inserted into the middle of the GFs’ pulp dermis and 10 μL of test material was delivered. GFs from each chicken were then collected at different time points post GF-pulp injections (p.i.) as indicated for each trial. Collected GF were used for same-day preparation of pulp cell suspensions or flash-frozen in liquid nitrogen and stored at −80 °C for later cytokine expression analyses.

#### Blood Sampling

Blood was collected (0.8 mL) from the wing vein of each chicken using heparinized 1 mL syringes with 25-gauge × 1-inch needles (Becton Dickinson, Franklin Lakes, NJ, USA). Gentle pressure was applied at the blood collection site to stop the bleeding entirely before returning the chickens to their pens. Additionally, the wing veins at the right and left side of the chicken were alternated between blood collection time points. Whole blood was used for direct immunofluorescent staining of whole blood cell suspensions and cell population analysis by FACS. Plasma was isolated and stored at −80 °C until later assessment of AGP-1 or relative plasma levels of antibody-specific IgM, IgG, and IgA by ELISA.

### 2.4. Preparation of GF-Pulp and Blood Cell Suspensions and Immunofluorescent (IF) Staining Procedures

GF-pulp cell suspensions were prepared at each time point, as described previously [17]. Briefly, for each GF, the pulp (living tissue) was removed from the feather sheath and placed in a 1.5 mL tube containing 0.5 mL of 0.1% collagenase/dispase (Collagenase Type IV, Life Technologies, Carlsbad, CA, USA; Dispase II, Sigma-Aldrich, St. Louis, MO, USA), and incubated at 37 °C for 15 min. The single-cell pulp suspensions were then prepared using a 60 μm nylon mesh filter while simultaneously adding ample ice-cold PBS. The pulp cell suspensions were further processed by washing and resuspending and kept on ice until the immunofluorescent staining procedure.

For blood cell suspension preparation, 20 μL of blood was added to 980 μL of PBS+ (PBS, 0.1% sodium azide, and 1% bovine serum albumin (BSA); VWR, Radnor, PA, USA), making a 1:50 dilution.

A panel of fluorescently labeled mouse monoclonal antibodies specific for chicken (mac) leukocyte cell surface molecules (Southern Biotechnology Associates, Inc., Birmingham, AL, USA) were used in a 2- or 3-color direct immunofluorescent staining procedure as described in [17]. Fluorescently labeled mac antibodies used for IF staining of pulp- and blood-cell suspensions included CD45-SPRD (total leukocytes), T cell receptor (TCR)-1-PE (γδ T cells), CT4-FITC (CD4^+^ T cells), CT8-PE (CD8α^+^ T cells), KUL-01-FITC, (macrophages/monocytes), and Bu-1-PE (B cells). Additionally, mac CD41/61-FITC (Bio-Rad, Hercules, CA, USA) was included for blood cell suspension to identify thrombocytes. Staining controls were as described [17].

For IF staining of pulp cell suspension, 50 μL of cells were incubated for 30 min at 4 °C with 50 μL of antibody cocktails, washed twice in PBS+, and 200 μL of the final suspensions was subjected to cell population analysis by FACS. For IF staining of blood cell suspensions, 50 µL cells were incubated for 45 min at 4 °C with 50 μL of antibody cocktails. After the incubation, samples were then further diluted by adding 300 μL of PBS+, yielding an overall 1:400 whole blood dilution for FACS analysis.

The BD C6 Accuri flow cytometer (Becton Dickinson, San Jose, CA, USA) and FlowJo software v10.5.2 (Flow Jo, LLC, Ashland, OR, USA) were used for cell population analysis. Due to the lack of fluorescently labeled antibodies to identify chicken heterophils, this population was identified using the size (FSC) and internal complexity (SSC) characteristics of the CD45^+^ GF-pulp or the CD45^+^CD41/61^−^ blood cells [20]. Pulp data were expressed as the % of total pulp cells, whereas the whole blood data were calculated and expressed as cell concentrations (10^3^ cells/μL of blood) by multiplying the percentage of each leukocyte population in the blood cell suspensions by the concentration of total blood cells and dividing by 100. The total lymphocyte population was calculated by the addition of T and B cells.

### 2.5. RNA Isolation, Quantification, cDNA Synthesis, and Relative Expression of Inflammatory Cytokines

From each sample and time point, the pulps of 2 frozen GF were removed from their sheath to isolate total RNA as previously described [17]. The quality and concentration of total RNA were determined using a NanoDrop (Thermo Fisher Scientific, Waltham, MA, USA). A high-capacity cDNA kit with MultiScribe Reverse Transcriptase was used according to the manufacturer’s protocol (Thermo Fisher Scientific) to transcribe RNA into cDNA. The cDNA samples were diluted to a working concentration of 10 ng/µL in nuclease-free water. Samples were then stored at −20 °C until relative gene expression analysis of inflammatory cytokines [17].

The primer and probes for the genes assessed in this study, including interleukin (IL) 1 beta (IL1B), IL6, and IL8 (CXCL8), and the 28S reference gene, were used to conduct real-time quantitative PCR (qRT-PCR) as previously described [16,19]. Data were expressed as 40^−ΔCt^ [21,22].

### 2.6. Reactive Oxygen Species Generation Assay

Pulp cell suspensions were used to assess reactive oxygen species (ROS) generation utilizing 2′,7′-dichlorofluorescin diacetate (DCFDA, Sigma, St. Louis, MO, USA) with a kinetic fluorescence assay, as described previously [23]. The fluorescence microplate reader (Synergy HTX; BioTek, Winooski, VT, USA) with a 485 nm excitation wavelength and a 530 nm emission wavelength was set at 37 °C to measure the generation of fluorescence for 1.0 h at 10 min intervals via kinetic read. Each plate included fluorescence and auto-fluorescence controls, and the Gen5 software 2.07 (BioTek, Winooski, VT, USA) was used to acquire the data. The relative amount of ROS generation was described by the relationship between fluorescence (a.u.) and time in a line of best fit, which was calculated for each sample and expressed as the slope of the best-fit equation [16].

### 2.7. Plasma Alpha-1 Acid Glycoprotein-1 Assay

The plasma concentrations (mg/mL) of alpha-1 acid glycoprotein-1 (AGP-1) in the blood samples collected before (0 h) and at 6 and 24 h post i.d. GF-pulp injection (p.i.) were evaluated. Chicken AGP-1 plasma levels were measured by ELISA following the manufacturer’s procedures (Abcam, Waltham, MA, USA).

### 2.8. Enzyme-Linked Immunosorbent Assay (ELISA) to Detect Plasma Levels of Salmonella Enteritidis-Specific Antibodies

The relative levels of SE-specific IgM, IgG, and IgA antibodies in the plasma of chickens were determined, as described previously [18]. Briefly, 100 µL of formalin-inactivated *Salmonella* Enteritidis bacteria in coating buffer (0.05 M carbonated buffer, pH 7.4; 1 × 10^7^ CFU/mL) was added to each well of a 96-well flat-bottomed plate (Thermo Scientific, Waltham, MA, USA, 262146). Plates were incubated at 37 °C for 2 h (Isotemp incubator, Fisher Scientific, Waltham, MA, USA), and then overnight at 4 °C. In the morning, plates were washed with a washing solution (50 mM Tris, 0.14 M NaCl, 0.05% Tween 20; pH 8.0) 3× for 2 min. Blocking buffer (50 mM Tris, 0.14 M NaCl, 1% BSA, pH 8.0; 200 µL/well) was then added to the plates. Plates were incubated for 30 min at room temperature, washed 3× for 2 min, and then assigned to IgM, IgG, or IgA detection. Plasma samples were thawed and diluted in sample diluent buffer (50 mM Tris, 0.14 M NaCl, 1% BSA, 0.05% Tween 20, pH 8.0) at 1/100, 1/1000, and 1/10,000 for IgA, IgM, and IgG detection, respectively. A standard curve from high to low concentration, which served as a positive control, was developed using a pool of plasma samples with high SE-specific antibody levels, diluted in a 6-step serial dilution. Each plate had additional controls (100 µL/well), including triplicate wells with the highest concentration of pool sample and diluent instead of sample to serve as blank and non-specific binding (NSB) controls. Plates were then incubated at 37 °C for 2 h, followed by a wash step (3× for 2 min). Detection antibodies (horse-radish-peroxidase (HRP)-conjugated; Bethyl Laboratories, Montgomery, TX, USA), prepared in sample diluent, included affinity purified goat anti-chicken (gac) IgG-Ig γ-heavy chain (1/20,000), IgM (1/20,000), and IgA (1/10,000). Detection antibodies were added (100 µL/well) to their respective plates, except for the blank controls, to which unconjugated gac IgG, IgM, and IgA were added instead. To quantify the binding of HRP conjugated detection antibodies, 100 µL TMB substrate was added to each well. Then, a 15 min incubation at 37 °C allowed for a color change reaction, which was stopped by adding 100 µL/well of 2M sulfuric acid solution. Using a 96-well spectrophotometer (ELx 800, BioTek, Winooski, VT, USA), the colorogenic substrate reaction of each well was measured in absorbance units (a.u.) at 450 nm. In each ELISA, the standard curve of each plate provided a line equation to adjust for inter-assay plate variation and evaluate the relative levels of IgM, IgG, and IgA antibody levels in chicken plasma.

### 2.9. Statistical Analysis

Sigma Plot 15.0 Software (Systat Software, Inc., San Jose, CA, USA) was used for all statistical analyses [17,19]. For each trial, data from each treatment group were analyzed separately, due to interactions involving treatment (e.g., LPS vs. PBS) and 2-way analysis of variance (ANOVA) was conducted to determine the effects of line (MRB, LWCR, and HWCR), time, and line-by-time interactions on all measurements except the blood, AGP-1, and antibody data, which were analyzed by 2-way repeated measures (RM) ANOVA. The Holm–Sidak method was applied as appropriate for multiple mean comparisons, and for all cases, statistical significance was set at *p* ≤ 0.05.

## 3. Results

### 3.1. Trial 1—Local and Systemic Inflammatory Responses to i.d. GF-Pulp Injections of Lipopolysaccharide (LPS) in 10-Week-Old Male Broiler Breeders from the MRB, LWCR, and HWCR Lines

#### 3.1.1. Leukocyte Population Response Profiles in GF-Pulps and Peripheral Blood Following i.d. GF-Pulp Injection of LPS

Statistical analysis revealed the main effects of line and time for all leukocyte populations in the GF-pulp and blood, except for monocytes in the blood, where a line-by-time interaction was observed.

LPS injection in GF-pulp resulted in elevated (*p* < 0.001) heterophil infiltration (% pulp cells) by 6 h post-pulp injection (p.i.), followed by a decline to still above pre-injection levels by 24 h p.i. (Figure 1A). There were no differences between the lines in the heterophil response. In the peripheral blood, concentrations (10^3^ cells/μL) of heterophils were also impacted by i.d. GF-pulp injection of LPS (main effects of time; *p* < 0.001), with maximal heterophil concentrations at 6 h p.i., followed by a return to pre-injection concentrations at 24 h p.i. (Figure 1B).

The i.d. GF-pulp injections of LPS caused a gradual influx of macrophages into the pulp, with elevated levels (*p* < 0.001) at 6 h p.i. and a further increase to maximal levels (*p* < 0.001) at 24 h p.i. There were no line differences in the GF-pulp infiltration of monocyte/macrophages in response to LPS injections (Figure 1C). In the blood, the monocyte concentrations changed over time in MRB and LWCR broilers, but not in HWCR broilers. In MRB and LWCR broilers, the LPS pulp injection resulted in increased monocyte concentrations at 6 h p.i., followed by a decline to near pre-injection levels by 24 h p.i. (Figure 1D). There were no line differences in monocyte concentrations at any of the time points examined.

For lymphocyte levels in the GF-pulp, there was no main effect of time, but a main effect of line (*p* = 0.035), where the LWCR was different from HWCR, but the MRB line was not different from the other lines (Figure 1E). In the blood, statistical analysis revealed a time main effect (*p* < 0.001), where lymphocyte concentrations dropped to lowest at 6 h p.i. and then returned to near pre-injection concentrations at 24 h p.i. (Figure 1F). While not significant, a return to baseline levels of lymphocytes at 24 h p.i. was most evident in LWCR broilers. In both the GF-pulps and blood, individual T- and B- cells followed the same pattern of changes over time as shown for total lymphocytes.

#### 3.1.2. Cytokine mRNA Expression in GF-Pulps Following i.d. GF-Pulp Injection of LPS

There were no line differences in the relative mRNA expression of IL-1β, IL-6, and IL-8 in response to i.d. GF-pulp injection (Figure 2). However, independent of the line, i.d. the injection of GF-pulps with LPS resulted in an increase (*p* < 0.001) in mRNA expression of IL-1β and IL-8 at 6 h and 24 h p.i., and no change in IL-6 (Figure 2).

#### 3.1.3. Reactive Oxygen Species (ROS) Generation in GF-Pulps and Alpha-1-Acid Glycoprotein-1 (AGP-1) Plasma Concentrations Following i.d. GF-Pulp Injection of LPS

There were no line differences in ROS generation in LPS-injected GF-pulps. In the MRB, LWCR, and HWCR lines, ROS generation (slope; FU × 1000/min) in GF-pulps increased (*p* < 0.001) in response to peak levels at 6 h, before returning to pre-injection levels at 24 h post-LPS injection into GF-pulps (Figure 3A). Independent of the line, the concentrations of AGP-1 in the plasma increased (*p* = 0.003) in response to LPS, reaching maximal levels at 24 h p.i. (Figure 3B).

### 3.2. Trial 2—Local and Systemic Inflammatory Responses to i.d. GF-Pulp Injections of Peptidoglycan (PGN) in 11-Week-Old Male Broiler Breeders from the MRB, LWCR, and HWCR Lines

#### 3.2.1. Leukocyte Population Responses Profiles in GF-Pulps and Peripheral Blood Following i.d. GF-Pulp Injection of PGN

In response to the i.d. GF-pulp injection of PGN, heterophil pulp-infiltration levels peaked (*p* < 0.001) by 6 h p.i., and remained elevated at 24 and 48 h, before declining to pre-injection levels by 72 h p.i., independent of the line (Figure 4A). There were no line differences in heterophil levels in response to PGN. In the blood, the i.d. GF-pulp injection of PGN influenced the concentrations of heterophils over time (*p* < 0.001). Independent of the line, heterophils reached peak concentrations at 6 h p.i. and returned to baseline levels at 24 h p.i. (Figure 4B). There were no differences in heterophil concentrations between the broiler lines.

There was a main effect of time (*p* < 0.014) on macrophage infiltration into PGN-injected GF-pulps, with macrophages reaching peak levels (*p* < 0.014) at 6 h p.i., followed by a decline to pre-injection levels thereafter (Figure 4C). The i.d. PGN injections did not affect circulating concentrations of monocytes (Figure 4D). There were no line differences in monocyte/macrophage levels in blood/GF-pulps, respectively.

There was a main effect of line (*p* < 0.001) and time (*p* < 0.001) on lymphocyte levels in PGN-injected GF-pulps (Figure 4E). Independent of the line, lymphocyte levels in the GF-pulp were elevated (*p* < 0.001) at 24 h p.i., reached peak (*p* < 0.001) levels at 48 h p.i., and then declined (*p* < 0.001) to above pre-injection levels at 72 h p.i. Lymphocyte infiltration into the GF-pulp was not different between MRB and LWCR lines, but was greater (*p* < 0.001) in both lines compared to the HWCR line (Figure 4E). In the blood, a main effect of time (*p* < 0.019) was observed in response to PGN GF-pulp injections, where lymphocyte concentrations gradually declined to their lowest levels at 24 h p.i. (Figure 4F). There were no differences in lymphocyte concentrations between the lines.

In PGN-injected GF-pulps, the higher lymphocyte infiltration in the LWCR and MRB lines compared to the HWCR line was due to the greater infiltration of both T- and B-cells, an observation also reflected by line-by-time interactions (T cells *p* = 0.049; B cells *p* = 0.014). In the MRB line, T cell recruitment levels peaked (*p* < 0.001) at 24 h and remained sustained at 48 h before returning to pre-injection levels (*p* < 0.001) by 72 h p.i., whereas, for the LWCR line, T cell levels peaked (*p* < 0.001) at 48 h and remained (*p* = 0.017) above pre-injection levels at 72 h. The T cell levels did not significantly change over time in the HWCR broilers. Additionally, at 24 h p.i., T cell levels were higher in the MRB line than in the LWCR and HWCR lines, while at 48 h p.i., T cell levels were higher in the LWCR line than in the HWCR line, and the MRB line had intermediate levels (Figure 5A). Similarly, in the MRB line, B cells reached peak levels (*p* = 0.014) at 24 h and remained elevated at 72 h p.i., whereas for the LWCR line, B cell levels peaked (*p* = 0.014) later at 48 and 72 h p.i. and did not change over time in the HWCR line. At 24 h p.i., B cells in the GF-pulp were higher (*p* = 0.014) in the MRB line than in the HWCR and LWCR lines. At 72 h, B cell levels were higher in the LWCR line compared to the HWCR line, with the MRB line having intermediate levels (Figure 5B).

In the blood, only a main effect of time (*p* = 0.002) was observed. Following GF-pulp injections of PGN, both T and B cell concentrations in the blood declined (*p* = 0.002) to the lowest levels at 24 h (Figure 5C,D).

#### 3.2.2. Cytokine mRNA-Expression in GF-Pulps Following i.d. GF-Pulp Injection of PGN

There was a main effect of time (*p* < 0.001) on the relative mRNA expression of IL8 but not for IL-1β and IL-6. In response to i.d. GF-pulp injection of PGN, relative IL-8 mRNA expression increased (*p* < 0.001) to maximal levels at 6 h, and remained elevated above pre-injection levels at 24, 48, and 72 h p.i. (Figure 6). There were no line differences in the relative mRNA expression for these cytokines.

#### 3.2.3. Plasma Alpha-1-Acid Glycoprotein-1 (AGP-1) Concentrations Before and After i.d. GF-Pulp Injection of PGN

There were no line or time differences in the concentrations of plasma AGP-1 in response to i.d. PGN GF-pulp injection, although the effect of time approached significance (*p* = 0.08) with the AGP-1 concentration tending to increase by 24 h in MRB and LWCR broilers.

### 3.3. Trial 3—Local and Systemic Innate and Adaptive Immune Responses to a First i.d. GF-Pulp Immunization with a Formalin-Killed Salmonella Enteritidis Vaccine (SEV) in 14-Week-Old Broiler Breeder Pullets from the MRB, LWCR, and HWCR Lines

#### 3.3.1. Leukocyte Population Profiles in GF-Pulps Before and After i.d. GF-Pulp Injection of SEV

In response to i.d. GF-pulp injections of SEV, the statistical analysis revealed time main effects for all leukocytes in the GF-pulp and blood, except for GF-pulp lymphocytes.

The first immunization with SEV by i.d. GF-pulp injection stimulated the highest (*p* < 0.001) heterophil infiltration levels at 6 h p.i., followed by a drop (*p* < 0.001) at 24 h, and a gradual return to pre-injection levels by 72 h p.i. (Figure 7A). In the blood, the concentration of circulating heterophils dropped (*p* < 0.001) to its lowest level at 6 h post GF-pulp injection of SEV, then increased (*p* < 0.001) to its highest level at 24 h p.i., followed by another drop (*p* < 0.001) at 48 h, before returning to near pre-injection concentrations at 72 h p.i. (Figure 7B).

Independent of line of broilers, the macrophage levels in the GF-pulp and monocyte concentrations in the blood changed over time (*p* < 0.001) post GF-pulp injection of SEV. In the GF-pulps, macrophages reached maximal levels (*p* < 0.001) at 24 h p.i. and gradually declined thereafter to pre-injection levels at 72 h (Figure 7C). The blood monocyte concentrations gradually declined (*p* < 0.001) to the lowest levels at 24 h p.i., followed by a rapid increase (*p* < 0.001) at 48 h, and declined once more (*p* < 0.001) to the low levels observed at 24 h p.i. (Figure 7D).

While there was no time main effect on lymphocyte levels in SEV-injected GF-pulps, lymphocyte levels differed between the lines (*p* < 0.001). Overall, lymphocyte levels in the GF-pulp were higher (*p* < 0.001) with the MRB and LWCR lines than with the HWCR line (Figure 7E). There was a line-by-time interaction in the blood lymphocyte concentrations (Figure 7F). While in MRB and LWCR broilers, blood lymphocyte concentrations did not change over time post-SEV injection, in HWCR broilers, the lymphocyte concentrations dropped (p < 0.001) to their lowest levels 6 h p.i., then increased over 24–48 h p.i., but by 72 h, they remained below the baseline. At 6 h p.i., the lymphocyte concentrations were higher (*p* = 0.05) in the MRB broilers compared to the HWCR broilers, with intermediate levels in the LWCR broilers (Figure 7F).

Further examination of the lymphocyte subpopulations affected by the i.d. SEV GF-pulp injection revealed no time or line main effects on T cells in GF-pulps, but levels of B cells were overall higher in MRB and LWCR than in HWCR broilers, and GF-pulp B cell levels changed over time independent of the line (Figure 8A,B). In the blood, T cell concentrations dropped (*p* < 0.001) to the lowest at 24 h p.i. and then returned (*p* < 0.001) to pre-injection concentrations at 48 h and remained at these levels at 72 h p.i. (Figure 8C), whereas B cell concentrations dropped greatly post GF-pulp injection and remained below pre-injection levels throughout the 72 h examination period (Figure 8D).

#### 3.3.2. *Salmonella* Enteritidis-Specific IgM, IgG, and IgA Antibody Levels in Plasma Before and After Primary GF-Pulp Immunization with Formalin-Killed *Salmonella* Enteritidis Vaccine (SEV)

Independent of the line, relative levels (a.u.) of SEV-specific IgM, IgG, and IgA antibodies in plasma changed with time post-SEV injection into GF-pulps (Figure 9). For all lines, levels of SEV-specific IgM gradually increased, reaching peak levels (p < 0.001) on day 10, remained near this level by 14 d, and then gradually decreased to above pre-injection levels (Figure 9A). Relative levels of SEV-specific plasma IgG increased in response to i.d. GF-pulp injections of SEV in all the lines, with peak levels (*p* < 0.001) observed at 7 d that then declined (*p* = 0.022) at 10 d and remained near this level at 14 and 21 d before increasing (*p* < 0.001) again to near peak levels at 28 d p.i. (Figure 9B). Independent of the line, relative plasma levels of SEV-specific IgA sharply increased (*p* < 0.001) to their highest levels at 5 d, remained at these levels at 7 d, and then declined (*p* = 0.009) to above pre-injection levels at 10 d, and remained at this level throughout the 28-day examination period. (Figure 9C). There were no differences between lines for the circulating SEV-specific IgM, IgG, and IgA levels. Hence, data were pooled across lines in Figure 9. Independent of the line, SEV-specific antibodies did not change over time post-PBS injection, and there were no differences in antibody levels between lines.

### 3.4. Leukocyte Response Profiles in GF-Pulps and Peripheral Blood Following Intradermal Pulp Injections of Endotoxin-Free Phosphate-Buffered Saline (PBS-Vehicle) in Broiler Breeders from the MRB, LWCR, and HWCR Lines from Three Trials

A four-way ANOVA was conducted for trial, sex, line, and time, and in the absence of interactions, data were pooled across trials (Trials 1, 2, and 3), sex, and line and analyzed by one-way ANOVA to assess the effects of time.

Following i.d. GF-pulp injections with PBS, heterophils and monocyte/macrophages infiltrated into the GF-pulps, whereby levels of both types of leukocytes peaked at 6 h p.i., returned to pre-injection levels at 24 h p.i., and remained near these levels thereafter (*p* < 0.001). There was a similar increase in heterophils and monocytes in the blood, with concentrations reaching highest at 6 h p.i., followed by a return to near pre-injection concentrations at 24 h p.i. (Table 1).

PBS administration by i.d. GF-pulp injections did not result in the recruitment of lymphocytes into the injected tissue and did not alter lymphocyte concentrations in the blood (Table 1).

## 4. Discussion

A primary goal of breeding broilers with improved water efficiency is to contribute to the sustainable use of water resources in poultry farming while providing high-quality products for consumers. It is well known that a poorly balanced immune system in broiler breeders and their offspring can increase susceptibility to infectious diseases, impacting animal health, well-being, and performance. Therefore, this longitudinal study aimed to better understand how genetic selection based on WCR influenced innate and adaptive immune system capabilities in broiler breeder lines. Specifically, to evaluate cellular and humoral immune responses, broiler breeders from the MRB control population, LWCR (high water efficiency) and HWCR (low water efficiency) lines received i.d. GF-pulp injections of MAMPs from Gram-negative (LPS) and Gram-positive (PGN) bacteria, or a first i.d. GF-pulp immunization with a killed *Salmonella* Enteritidis vaccine (SEV). Based on these results, the LCWR broilers were found to be equal to the MRB controls in their ability to respond to the bacterial MAMPs and SEV, demonstrating that selection for LWCR did not affect the innate and adaptive defenses of broilers to bacterial challenges.

### 4.1. Assessment of Local and Systemic Inflammatory Responses to i.d. GF-Pulp Injections of Lipopolysaccharide (LPS) in 10-Week-Old Male Broiler Breeders from the MRB, LWCR, and HWCR Lines

The i.d. injection of GF-pulps with LPS in the MRB, LWCR, and HWCR lines resulted in the successful recognition of this MAMP by the broiler breeders’ immune system, triggering an organized local, acute inflammatory response, fully in line with the temporal changes in leukocyte profiles reported following i.d. LPS injection in commercial broilers and in juveniles from egg-layer lines [16,19]. Specifically, in all three lines, there was a rapid infiltration of heterophils in the GF-pulps, with peak levels observed at 6 h p.i., declining substantially by 24 h p.i., and remaining above baseline levels at 72 h p.i. This heterophil influx was accompanied by greatly elevated ROS generation, reaching peak levels at 6 h p.i., and infiltration of monocytes/macrophages into the GF-pulps, with proportions increasing at 6 h, reaching the highest levels at 24 h p.i., and remaining elevated thereafter. These temporal changes in phagocyte presence post-LPS injection reflect a responsive innate immune system. As expected, heterophils were the first to infiltrate the site of LPS injection. They are highly effective phagocytes and generators of ROS and play a crucial role in the removal/elimination of extracellular microbes. As heterophils are short-lived, the observed concomitant recruitment of monocytes/macrophages, and the sustained presence of macrophages observed in all three lines, also reflect the broiler lines’ ability to mount a well-coordinated and robust acute inflammatory response. Macrophages are not only excellent phagocytes, but once activated with LPS, they produce pro-inflammatory cytokines, including chemokine IL-8 and pro-inflammatory cytokines IL-1β and IL-6. In all three lines examined, qRT-PCR analysis of GF-pulps revealed an increase in the relative mRNA expression of IL-1β and IL-8 at 6 h p.i., with levels remaining elevated at 24 h p.i., suggesting that the LPS injection produced a local/tissue cellular inflammatory environment that persisted over these time points. Overall, these results align with the observations highlighted in this study’s phagocyte pulp-infiltration profiles, as IL-1β is an essential mediator of inflammation, and IL-8 is a crucial stimulator of heterophil recruitment [16,24]. Following the i.d. GF-pulp administration of LPS, another effect was observed in the recruitment of lymphocytes, with the LWCR line recruiting higher levels into the GF-pulp than the HWCR line, while the MRB line had intermediate levels.

The local inflammatory activity generated with LPS in GF-pulps also stimulated systemic inflammatory activity as measured in the peripheral blood circulation. Specifically, concentrations of heterophils were increased at 6 h p.i. in all lines, and those of monocytes also at 6 h, but only in the MRB and the LWCR lines. Furthermore, lymphocyte concentrations dropped at 6 h in all three lines. In all lines, the levels of these leukocytes returned to near baseline by 24 h. Additionally, the AGP-1 concentrations in the plasma increased to the highest in all lines at 24 h post-i.d. GF-pulp injection of LPS; further supporting that selection for improved water efficiency (LWCR) did not affect the production of this protective and regulatory acute phase protein. These changes in circulating leukocyte and AGP-1 concentrations post-i.d. LPS injection also followed the general patterns reported for commercial broilers and juvenile egg-type chickens receiving i.d. GF-pulp injections with the same dose of LPS [16,17,19].

Taken together, the MRB and LWCR lines were shown to be similarly capable of responding to LPS with an appropriate acute inflammatory response in a complex tissue and initiating systemic changes in blood leukocyte and AGP-1 concentrations to support the local inflammatory activity. The ability to mount a robust acute inflammatory response to this MAMP reflects the lines’ ability to respond to infections with Gram-negative bacteria, and importantly, the divergent selection pressure on improving the WCR did not impair this cellular, innate defense mechanism in the LWCR line.

### 4.2. Local and Systemic Inflammatory Responses to i.d. GF-Pulp Injections of Peptidoglycan (PGN) in 11-Week-Old Male Broiler Breeders from the MRB, LWCR, and HWCR Lines

The assessment of the in vivo immunostimulatory effects of MAMPs like peptidoglycan (PGN), a cell wall component found in Gram-positive bacteria, is an underdeveloped area of research; although, Byrne and Erf (2024) recently reported a lymphocyte-dominated GF-pulp infiltration in response to various doses (0.05, 0.5, and 5 µg/GF) of PGN administered by i.d. injections in juvenile egg-type chickens [19]. To study the responses to i.d. PGN administration in the MRB, LWCR, and HWCR broiler breeder lines, 1 µg PGN/GF was used, as Byrne and Erf (2024) reported no differences in the inflammatory responses generated by 0.5 or 5 µg/GF [19].

In general, the changes in local leukocyte profiles initiated by PGN injection of GF-pulps in the three broiler lines aligned well with the time course, types, and levels of leukocyte infiltration into injected GF-pulps and with the accompanying changes in blood leukocyte concentrations observed in similar age egg-type chickens [19].

In all three broiler lines, i.d. GF-pulp injections with PGN resulted in an influx of both heterophils and monocytes/macrophages into GF-pulps with maximal levels at 6 h and a return to baseline levels by 72 h and 24 h p.i., respectively, while in the blood, concentrations of heterophils were elevated at 6 h and monocyte concentrations did not change. Moreover, as with the egg-type chickens, lymphocytes were the dominant cell type recruited into injected GF-pulps, reaching greatly elevated levels at 24 and 48 h, and remained well above pre-injection levels at 72 h p.i. Similarly, there was a drop in the concentrations of circulating lymphocytes by 24 h, which was also observed in egg-type chickens [19]. Based on a side-by-side comparison of the above described LPS- and PGN-stimulated inflammatory responses, levels of heterophils and monocytes/macrophages in PGN-injected GF-pulps were much lower, and those of lymphocytes much higher, than in LPS-injected GF-pulps. This was also reported for egg-type chickens and further emphasizes the divergent response profiles initiated by LPS and PGN in chickens [19].

A comparison of the PGN-stimulated responses between the MRB, LWCR, and HWCR lines, however, revealed line differences in the levels of lymphocytes present in PGN-injected GF-pulps. Specifically, the levels of lymphocytes recruited were nearly twice as high in the MRB and LWCR lines than in the HWCR line, whereby the lower lymphocyte levels in HWCR broilers were primarily due to the much lower levels of infiltrating B cells. While the mechanism underlying lymphocyte recruitment in response to PGN is not yet clear, the MRB and LWCR lines are equally effective in recruiting lymphocytes, and, hence, divergent selection for LWCR did not affect the capability of the LWCR broilers to respond to PGN and, more broadly, potentially to infections with Gram-positive bacteria. There were also no line differences in the mRNA expression profiles for IL-1β, IL-6, and IL-8, or in the plasma AGP-1 concentrations, although the IL-1β mRNA expression in the pulp and the plasma AGP-1 concentrations increased slightly over time in the MRB and LWCR broilers, but not in the HWCR broilers.

Taken together, the MRB and LWCR lines were shown to be similarly capable of responding to PGN with an appropriate, lymphocyte-dominated inflammatory response at the site of injection. While the mechanisms underlying this response are not well understood, it indicates that the genetic selection for LWCR did not negatively impact the broilers’ ability to respond to this MAMP that is abundantly present in Gram-positive bacterial pathogens, such as *Clostridium perfringens*, *Staphylococcus* spp., and *Enterococcus* spp., which continue to pose a threat to broiler health in poultry production [25,26,27,28,29].

### 4.3. Local and Systemic Innate and Adaptive Immune Responses to a First i.d. GF-Pulp Immunization with a Formalin-Killed Salmonella Enteritidis Vaccine (SEV) in 14-Week-Old Broiler Breeder Pullets from the MRB, LWCR, and HWCR Lines

Immunization with a killed *Salmonella* Enteritidis vaccine (SEV) in naïve broiler pullets from the MRB, LWCR, and HWCR lines by i.d. GF-pulp injections resulted in the infiltration of leukocytes from the blood into the injected tissues as well as in a robust humoral, SEV-specific IgM, IgG, and IgA antibody production in the broiler breeder lines.

In all three lines, the immune system identified the SEV as foreign, stimulating an organized and potent local inflammatory response dominated by the GF-pulp infiltration of phagocytes. The recruitment of heterophils reached maximal levels at 6 h and gradually declined to pre-injection levels by 72 h p.i., and monocyte/macrophage infiltration increased to highest levels at 24 h and returned to pre-injection levels at 72 h p.i. In the blood, independent of the line, the i.d. GF-pulp injections with SEV caused concentrations of heterophils and monocytes to drop at 6 h; heterophil concentrations then returned to near baseline levels by 24 h p.i., while the concentrations of monocyte remained low thereafter.

The phagocyte recruitment in response to i.d. GF-pulp injections with SEV follows a similar trend to that observed in egg-type pullets subjected to i.d. GF-pulp injections of autogenous *Salmonella*-killed vaccines where the dominant serovar in the vaccine formulation was Enteritidis [18]. The rapid influx of heterophils to the site of SEV administration corresponds with their role as the primary phagocytes to arrive at the site of infection. While short-lived, heterophils actively engulf, degranulate, and trap bacteria, greatly impacting the removal and clearance of the infection [30,31]. The prolonged presence of macrophages furthers this effort due to their heightened ability to respond to the various bacterial MAMPs by conducting increased phagocytic and bactericidal activities, antigen processing and presentation, and homeostatic regulation of the inflammatory response in the infected tissues [32,33]. The prominent participation of phagocytes in the response to i.d. GF-pulp injections of whole SEV bacteria in the three broiler lines suggests that the divergent selection based on WCR did not compromise their phagocyte activities in response to SEV and, likely, other Gram-negative bacteria.

Interestingly, following i.d. GF-pulp injections with SEV in the broiler breeder lines, lymphocyte levels in GF-pulps did not change significantly over time but were higher in the MRB and LWCR lines than in the HWCR line. These differences were mainly due to the greater infiltration of B cells in the MRB and LWCR broiler pullets. In the blood, the lymphocyte concentrations were similar overall in the three lines, but the lines responded differently post-SEV injection. While lymphocyte concentration did not change significantly in the MRB and LWCR broilers over the 72 h examination period, levels dropped at 6 h p.i. in HWCR broilers and did not recover by 72 h. However, in all three lines, a closer examination of T- and B-cell population changes over time post-SEV injection revealed a transient drop in T cell concentrations at 24 h p.i. and a sustained reduction in B cell concentrations from 6 to 72 h p.i.

Overall, the immunological responses of the LWCR broiler pullets were remarkably similar to those of the MRB controls in their ability to recruit leukocytes to the site of injection of killed SEV, and to stimulate local and systemic immune system activities to this bacterial vaccine. Hence, selection for improved water efficiency did not impair the LWCR broiler pullets’ ability to mount appropriate cellular immune responses, and as discussed below, stimulate a primary T cell-dependent, SEV-specific adaptive immune response.

The local inflammatory activities at the site of i.d. administration of SEV in the broiler breeder lines initiated processes leading to the activation of a SEV-specific adaptive immune response, as indicated by the appearance of SEV-specific IgM, IgG, and IgA antibodies in the peripheral blood. Independent of line, the relative plasma levels (a.u.) of SEV-specific IgM were elevated to maximal levels by 10 d p.i. and were followed by a gradual decline to above pre-injection levels by 28 d. The levels of plasma SEV-specific IgG increased gradually to maximal levels by 7 d, declined to above pre-injection levels between 10 and 21 d, and then increased again by 28 d p.i. The i.d. GF-pulp injection with vaccine also stimulated the production of plasma SEV-specific IgA, where levels were highest on 5 d, declined at 7 to 10 d, and remained above pre-injection levels thereafter. These antibody response profiles generated in response to the i.d. GF-pulp injection of SEV reflect characteristics of a T cell-dependent, primary humoral response, with a T helper cell-mediated isotype switch from SE-specific IgM to IgG and IgA. Moreover, the plasma SEV-specific IgM, IgG, and IgA antibody data observed in this broiler breeder study are consistent with the results from other longitudinal studies in specific pathogen-free and other egg-type chickens, where birds were challenged with *Salmonella* Typhimurium and immunized with autogenous killed *Salmonella* vaccines, respectively [18,34].

Overall, the i.d. GF-pulp injections with SEV led to a potent SEV-specific, T-dependent, primary humoral response in the three broiler breeder lines, further underlining that divergent selection for water efficiency did not negatively impact the LWCR broilers’ capabilities to mount effective cellular and humoral immune responses to a killed *Salmonella* vaccine.

### 4.4. Responses to Sterile, Endotoxin-Free PBS-Vehicle Injections

Independent of trial, sex, and line, all broilers responded similarly to the volume-matched, i.d. GF-pulp injection of the sterile, endotoxin-free PBS vehicle. The i.d. vehicle injections stimulated transient and low cellular inflammatory activities, characterized by elevated heterophil and monocyte/macrophage levels at 6 h, both at the site of injection and in the peripheral blood. These mild inflammatory responses to sterile injections have been repeatedly observed in broiler and egg-type chickens and reflect the inflammatory response to tissue injury [16,19].

## 5. Conclusions

In this longitudinal study, we demonstrated that broilers from the LWCR line and the MRB control line respond equally well to the bacterial MAMPs LPS and PGN and to a *Salmonella* bacterin i.d. injected into the pulp of growing feathers. Hence, the selection for more efficient water use in the LWCR broilers did not compromise their ability to mount robust innate and adaptive and cellular and humoral immune responses to these test materials from Gram-negative and Gram-positive bacteria. While the focus is on the more water-efficient LWCR line’s performance, it is important to mention that with all test materials, the lymphocyte recruitment into the injected GF-pulps was less prominent in the HWCR broilers than in the LWCR and MRB broilers. This finding shows that trait selection (e.g., WCR) can impact the immune system and that the GF-pulp bioassay used in this study is a practical tool for detecting immunological line differences in chickens. Additionally, this is the first comprehensive study on the temporal, qualitative, and quantitative characteristics of the local and systemic cellular inflammatory responses to LPS, PGN, and SEV in broiler breeders, which responded in a way remarkably similar, in all aspects examined, to that of egg-type chickens of similar age [18,19], despite being on different feeding programs at the time of the experiments (egg-type chickens were fed ad libitum; the broilers were on a breeder controlled feeding program). Overall, the results reported here provide foundational insights for more focused research on innate and adaptive immune responses to pathogens in meat- and egg-type chickens with applications in breeding programs that aim for strong and well-regulated immune system defenses.

## Figures and Tables

**Figure 1 vetsci-12-00279-f001:**
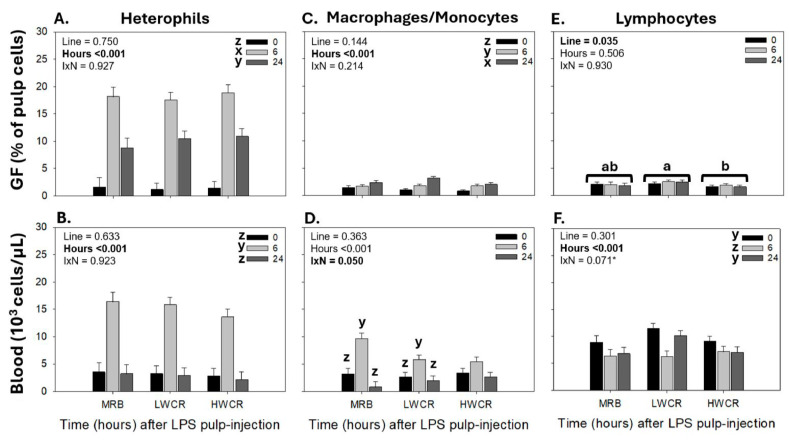
Changes in heterophil, monocyte/macrophage, and lymphocyte levels in growing feather (GF)-pulps (**A**,**C**,**E**) and whole blood (**B**,**D**,**F**) following intradermal pulp injection of lipopolysaccharide (LPS; *Salmonella* Typhimurium) in 10-week-old, male broiler breeders from the HWCR, LWCR, and MRB lines. GFs were injected with 10 μL of LPS (1 μg/GF; 12 GF/bird). GF-pulp and blood samples collected before (0 h) and at 6 h and 24 h p.i. were subjected to direct, 2–3 color, immunofluorescent staining to identify total leukocytes (CD45^+^), thrombocytes (CD41/61 in blood only), monocytes/macrophages, T cells, and B cells. Flow cytometry was used for leukocyte population analyses. The heterophil population was identified within the GF-pulp’s CD45^+^- and the blood’s CD45^+^CD41/61^−^ -population based on side- and forward-scatter characteristics. The T- and B-lymphocytes were added to calculate the lymphocyte population. To assess the effects of line, time, and line-by-time interactions (IxN), the GF-pulp leukocyte data were subjected to a two-way ANOVA, while the blood leukocyte data were analyzed using two-way repeated-measures (RM)-ANOVA. Data shown are mean leukocyte levels ± SEM; *n* = 6 broilers per line; a, b: line main effect means without a common letter are different; x, z: time main effect means, or time points within a line, without a common letter are different. * indicates a marginal effect. Statistical significance was set at *p* ≤ 0.05.

**Figure 2 vetsci-12-00279-f002:**
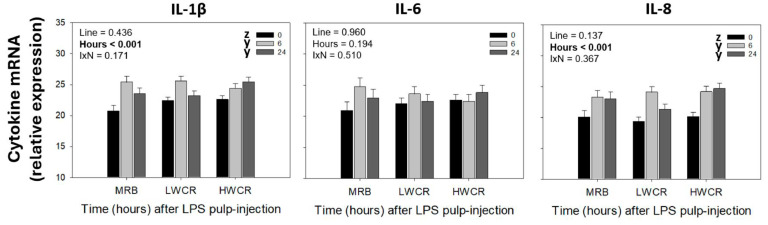
Relative mRNA expression of cytokines in growing feather (GF)-pulps following intradermal pulp injection of lipopolysaccharide (LPS; *Salmonella* Typhimurium) in 10-week-old, male broiler breeders from the MRB, LWCR, and HWCR lines. GF of broilers were i.d. injected with 10 μL of LPS (1 μg/GF; 12 GF/bird). GFs were collected before (0 h) and at 6 h and 24 h p.i. to isolate RNA from pulps for qRT-PCR to determine the relative mRNA expression (40^−ΔCt^) of interleukin 1-beta (IL-1β), IL-6, and IL-8 (CXCL8). To evaluate the effects of line, time, and line-by-time interactions (IxN), the GF-pulp gene expression data were analyzed by a two-way ANOVA. Data shown are mean ± SEM; *n* = 6 broilers per line; y, z: time main effect means without a common letter are different. Statistical significance was set at *p* ≤ 0.05.

**Figure 3 vetsci-12-00279-f003:**
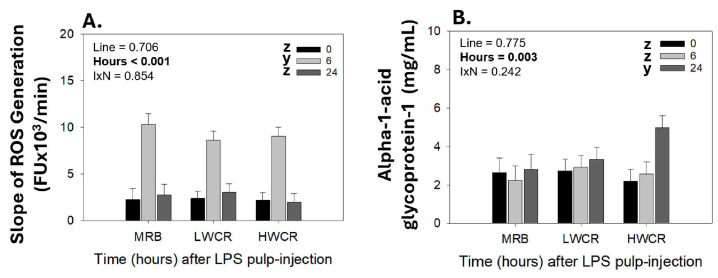
Reactive oxygen species (ROS) generation (**A**) and plasma concentrations of alpha-1-acid glycoprotein-1 (AGP-1) (**B**) following intradermal GF-Pulp injection of LPS in 10-week-old male broiler breeders from the MRB, LWCR, and HWCR lines. (**A**). GFs were collected before (0 h) and at 6 h and 24 h post LPS (*Salmonella* Typhimurium; 1 µg/GF; 12 GF/bird) injection to prepare pulp cell suspensions for use in a fluorescence kinetic assay to determine ROS generation. ROS generation was expressed as the slope of fluorescence emission (fluorescence units (FU) × 10^3^/min). (**B**). AGP-1 concentrations (mg/mL) were analyzed by ELISA in plasma samples isolated from heparinized blood collected before (0 h) and at 6 and 24 h after GF-pulp injection of LPS. To assess the effects of line, time, and line-by-time interactions (IxN), the ROS GF-pulp data were examined by a two-way ANOVA, while the plasma AGP-1 data were analyzed using two-way repeated-measures (RM)-ANOVA. Data shown are means ± SEM. For all assays, *n* = 6 broilers per line and treatment; y, z: time main effect means without a common letter are different (*p* ≤ 0.05).

**Figure 4 vetsci-12-00279-f004:**
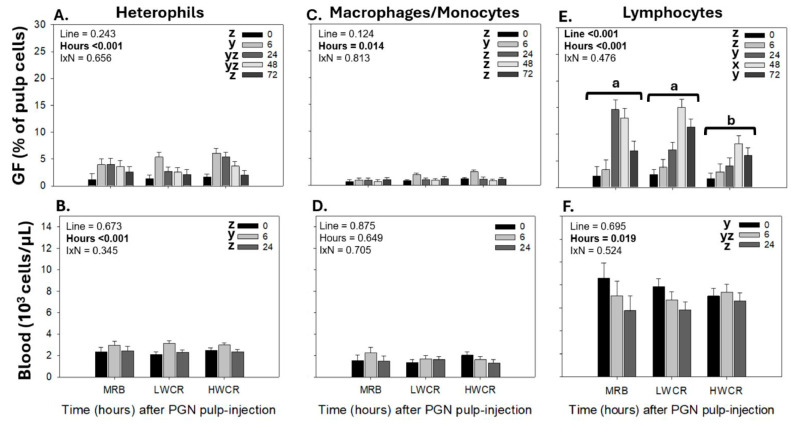
Changes in heterophil, monocyte/macrophage, and lymphocyte levels in growing feather (GF)-pulps (**A**,**C**,**E**) and whole blood (**B**,**D**,**F**) following intradermal pulp injection of peptidoglycan (PGN; *Staphylococcus aureus*) in 11-week-old male broiler breeders from the MRB, LWCR, and HWCR lines. GF-pulps were injected with 10 μL of PGN (1 μg/GF; 16 GF/bird). GF-pulp samples collected before (0 h) and at 6, 24, 48, and 72 h p.i., and blood samples collected at 0, 6, and 24 h p.i. were subjected to direct, 2–3 color, immunofluorescent staining to identify total leukocytes (CD45^+^), thrombocytes (CD41/61; only for blood), monocytes/macrophages, T cells, and B cells. Flow cytometry was used for leukocyte population analyses. The heterophil population was identified within the GF-pulp’s CD45^+^- and the blood’s CD45^+^CD41/61^−^ -population based on side- and forward-scatter characteristics. The T and B cells were added to calculate the lymphocyte population. To assess the effects of line, time, and line-by-time interactions (IxN), the GF-pulp leukocyte data were subjected to a two-way ANOVA, while the blood leukocyte data were analyzed using two-way repeated-measures (RM)-ANOVA. Data shown are mean leukocyte levels ± SEM; *n* = 6 broilers per line; a, b: line main effect means without a common letter are different; x, z: time main effect means without a common letter are different. Statistical significance was set at *p* ≤ 0.05.

**Figure 5 vetsci-12-00279-f005:**
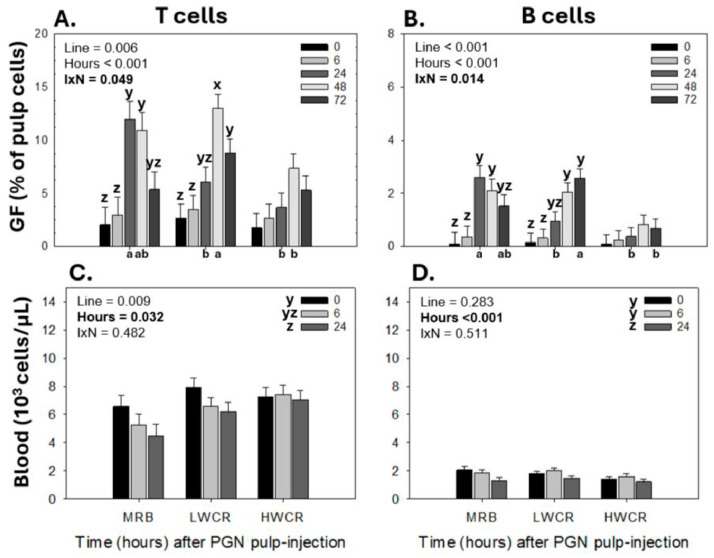
Changes in T and B cell levels in growing feather (GF)-pulps (**A**,**B**) and whole blood concentrations (**C**,**D**) following intradermal pulp injection of peptidoglycan (PGN; *Staphylococcus aureus*) in 11-week-old male broiler breeders from the MRB, LWCR, and HWCR lines. GF-pulps were i.d. injected with 10 μL of PGN (1 μg/GF; 16 GF/bird). GF-pulp samples collected before (0 h) and at 6, 24, 48, and 72 h p.i., and blood samples collected at 0, 6, and 24 h p.i., were processed for direct immunofluorescent staining (2–3 color) with fluorescently labeled mouse monoclonal antibodies to identify T- and B lymphocyte populations. Flow cytometry was used for lymphocyte population analyses. To assess the effects of line, time, and line-by-time interactions (IxN), the GF-pulp leukocyte data were subjected to a two-way ANOVA, while the blood leukocyte data were analyzed using two-way repeated measures (RM) ANOVA. Data shown are means ± SEM; *n* = 6 broilers per line; a, b: within a time point, means for each line without a common letter are different; x, z: time main effect means, or time points within a line, without a common letter are different. Statistical significance was set at *p* ≤ 0.05.

**Figure 6 vetsci-12-00279-f006:**
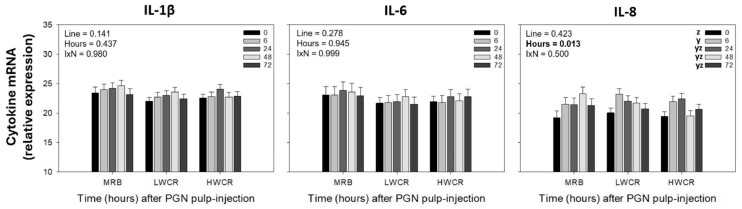
Changes in relative cytokine mRNA expression in growing feather (GF)-pulps following intradermal pulp injection of peptidoglycan (PGN; *Staphylococcus aureus*) in 11-week-old male broiler breeders from the MRB, LWCR, and HWCR lines. Broilers received 10 μL of PGN (1 μg/GF; 16 GF/bird). GF-pulp samples were collected before (0 h) and at 6, 24, 48, and 72 h p.i. for isolation of RNA to determine relative expression (40-ΔCt) of interleukin 1-beta (IL-1β), IL-6, and IL-8 mRNA by qRT-PCR. To evaluate the effects of line, time, and line-by-time interactions (IxN), the GF-pulp gene expression data were analyzed by a two-way ANOVA. Data shown are means ± SEM; *n* = 6 broilers per line; y, z: time main effect means without a common letter are different (*p* ≤ 0.05).

**Figure 7 vetsci-12-00279-f007:**
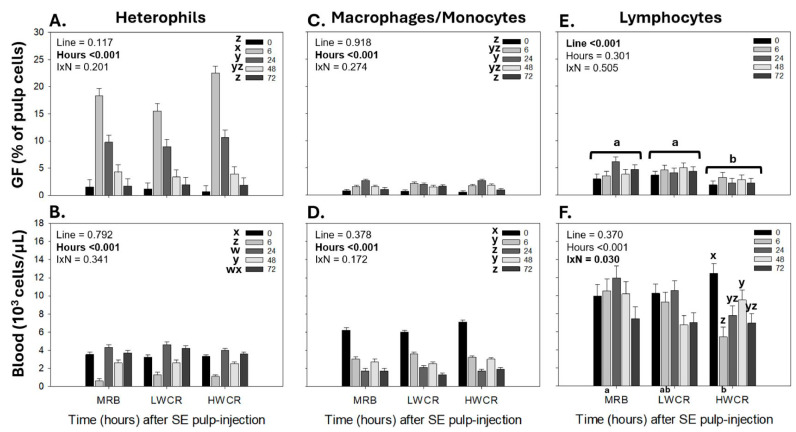
Changes in heterophil, monocyte/macrophage, and lymphocyte levels in growing feather (GF)-pulps (**A**,**C**,**E**) and whole blood (**B**,**D**,**F**) following a first intradermal pulp injection of a formalin-killed *Salmonella* Enteritidis vaccine (SEV) in 14-week-old broiler breeder pullets from the MRB, LWCR, and HWCR lines. Pullets were injected with 10 μL of SEV (10^8^ CFU/mL; 20 GF/bird). GF-pulp and blood samples collected before (0 h) and at 6, 24, 48, and 72 h p.i. were processed for direct immunofluorescent staining (2–3 color) with fluorescently labeled mouse monoclonal antibodies to identify total leukocytes (CD45^+^), thrombocytes (CD41/61 only for blood), monocytes/macrophages, T cells, and B cells. Flow cytometry was used for leukocyte population analyses. The heterophil population was identified within the GF-pulp’s CD45^+^- and the blood’s CD45^+^CD41/61^−^ -population based on side- and forward-scatter characteristics. The T and B cells were added to calculate the lymphocyte population. To assess the effects of line, time, and line-by-time interactions (IxN), the GF-pulp leukocyte data were subjected to a two-way ANOVA, while the blood leukocyte data were analyzed using two-way repeated measures (RM) ANOVA. Data shown are mean leukocyte levels ± SEM; *n* = 6 broilers per line; a, b: for each line, main time effect means without a common letter are different; a, b below *x*-axis: line means at a time point without a common letter are different; w, z: time main effect means, or time points within a line, without a common letter are different. Statistical significance was set at *p* ≤ 0.05.

**Figure 8 vetsci-12-00279-f008:**
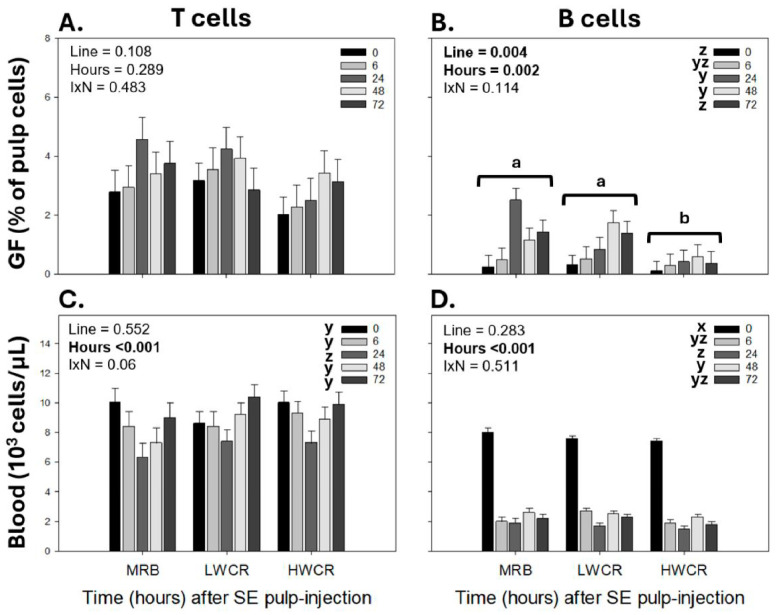
Changes in T and B cell levels in growing feather (GF)-pulps (**A**,**B**) and whole blood concentrations (**C**,**D**) following a first intradermal pulp injection of a formalin-killed *Salmonella* Enteritidis vaccine (SEV) in 14-week-old broiler breeder pullets from the MRB, LWCR, and HWCR lines. Pullets were injected with 10 μL of SEV (10^8^ CFU/mL; 20 GF/bird) into the pulp of GFs. GF-pulp and blood samples collected before (0 h) and at 6, 24, 48, and 72 h p.i. were processed for direct immunofluorescent staining (2–3 color) using fluorescently labeled mouse monoclonal antibodies to identify T- and B-lymphocyte populations. Flow cytometry was used for T and B lymphocyte population analyses. To assess the effects of line, time, and line-by-time interactions (IxN), the GF-pulp leukocyte data were subjected to a two-way ANOVA, while the blood leukocyte data were analyzed using two-way repeated measures (RM) ANOVA. Data shown are means ± SEM; *n* = 6 broilers per line; a, b: line main effect means without a common letter are different; x, z: time main effect means without a common letter are different. Statistical significance was set at *p* ≤ 0.05.

**Figure 9 vetsci-12-00279-f009:**
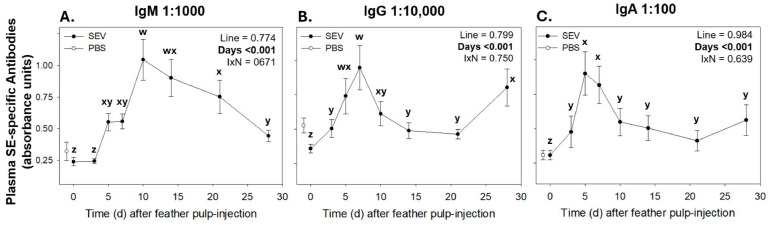
*Salmonella* Enteritidis-specific IgM (**A**), IgG (**B**), and IgA (**C**) levels in plasma after a primary immunization of formalin-killed *Salmonella* Enteritidis vaccine (SEV) or PBS vehicle into GF-pulps of 14-week-old broiler breeder pullets from the MRB, LWCR, and HWCR lines. Blood SEV-specific antibody levels in the plasma of 14-week-old broiler breeder pullets from the MRB, LWCR, and HWCR lines were measured after intradermal injection of SEV (10 μL/GF; 20 GF/bird) into the pulp of growing feathers. Plasma samples from heparinized blood (0.8 mL) collected before (0 d) and at 3, 5, 7, 10, 14, 21, and 28 d p.i. were subjected to ELISA. To evaluate the effects of line, time, and line-by-time interactions (IxN), the antibody data were analyzed using a two-way repeated measures (RM) ANOVA. SEV data shown are mean relative antibody levels ± SEM; *n* = 18 broilers (6 broilers/line). For PBS, data shown are the overall means (IgM, 0.28; IgG, 0.47; IgA, 0.27) across all lines and times; w, z: time point means without a common letter are different. Statistical significance was set at *p* ≤ 0.05.

**Table 1 vetsci-12-00279-t001:** Leukocyte profiles in GF-pulps and in peripheral blood following intradermal pulp injection of endotoxin-free phosphate-buffered saline (PBS) vehicle in MRB, LWCR, and HWCR broiler breeder chickens over three trials ^3^.

**GF-Pulp ^1^** (% Pulp Cells)	**Heterophils**	**Monocytes/** **Macrophages**	**Lymphocytes**
Time ^4^ (h)			
0 h	1.34 ± 0.09 z	0.76 ± 0.08 z	2.24 ± 0.28
6 h	7.45 ± 1.17 y	2.92 ± 0.45 y	2.54 ± 0.32
24 h	3.38 ± 0.38 z	1.41 ± 0.15 z	2.41 ± 0.20
48 h	2.90 ± 0.41 z	1.06 ± 0.13 z	2.56 ± 0.27
72 h	2.22 ± 0.34 z	1.14 ± 0.14 z	2.19 ± 0.23
Effects (*p*-value) ^5^			
Time	<0.001	<0.001	0.814
**Blood ^2^** (10^3^ cells/µL)			
Time (h)			
0 h	2.49 ± 0.21 z	3.72 ± 0.39 z	7.68 ± 0.57
6 h	3.61 ± 0.45 y	5.35 ± 0.48 y	9.39 ± 0.72
24 h	2.68 ± 0.24 yz	2.50 ± 0.33 z	7.87 ± 0.55
Effects (*p*-value)			
Time	<0.001	<0.001	0.108

^1^ Percentage of a leukocyte population in the pulp cell suspension. ^2^ Leukocyte concentration profiles in the peripheral blood. ^3^ In Trials 1–3, 3 broilers from each line received i.d. GF-pulp injections of sterile, endotoxin-free PBS vehicle instead of the Trial-specific test-materials [Trial 1, 10-week-old male broiler breeders; Trial 2, 11-week-old male broiler breeders; Trial 3, 14-week-old broiler breeder pullets]. Based on 4-way ANOVA results, data could be pooled across trials, lines, and sex, and a one-way ANOVA was conducted to determine the effect of PBS injection on leukocyte population profiles over time. ^4^ Time: GF-pulp samples were collected before (0 h) and at 6, 24, 48, and 72 h following GF-pulp injection (p.i.) of PBS (10 μL/GF) and heparinized blood samples before (0 h) and at 6 and 24 h p.i. Data shown are mean ± SEM; *n* = 18 broiler breeders (6 broiler/line). The Holm–Sidak multiple means comparisons were applied as appropriate. y, z: within GF-pulp or blood data, means without a common letter are different. ^5^ Statistical significance was set at *p* ≤ 0.05.

## Data Availability

The data presented in this study are available on request from the corresponding authors.

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
