# Peer review of "Selection for Improved Water Efficiency in Broiler Breeder Lines Does Not Negatively Impact Immune Response Capabilities to Gram− and Gram+ Bacterial Components and a Killed-Salmonella Enteritidis Vaccine"

_vetsci, 2025, doi:10.3390/vetsci12030279_

Round 1
Reviewer 1 Report
Comments and Suggestions for Authors
It is a good paper in writing, language and logic, which shall be acceptted
minior suggestions :
1, I.d. ( intradermal)infection? skin and musscle do not closely contact with gap, How to inject ?
2, Standard of LWCR and HWCR?
3, I do not understand GF-Pulp injection ?
4, cytokines ? if possible , Anti-inflammatory factors, such as IL-10 , shall be analyzed
5, Gram+ bacteria ? I do not find
6, Male Broiler Breeders were used , why not ot use female ?
Author Response
Comment 1. I.d. ( intradermal)infection? skin and musscle do not closely contact with gap, How to inject ?
Response 1: Thank you for your comment. To clarify:
- This study did not involve an infection but rather an injection. The killed Salmonella vaccine was administered intradermally as part of an immune response assessment, not to establish an infection model.
- The injected tissue (pulp) is not muscle but a skin structure with dermis and epidermis, vasculature and nerve endings.
- The growing feather (GF)-pulp is a skin derivative and a complex tissue that allows for repeated sampling of intradermally injected test material over time. This unique bioassay allows for the assessment of local immune responses in a minimally invasive manner. As explained in lines 87–95 and lines 171–181, the growing feather pulp provides a suitable window for evaluating tissue/cellular immune responses to different test materials, in this case, LPS, PGN, and a killed Salmonella
Comment 2, Standard of LWCR and HWCR?
Response 2: Thank you for your question. We assume that you are referring to the controls for the Low Water Conversion Ratio (LWCR) and High Water Conversion Ratio (HWCR) lines in our study. To clarify, the control for these lines is the Modern Random Breed (MRB) control line, which was detailed in lines 113-117 and referred to throughout the manuscript. The HWCR and LWCR lines were selected from the MRB control line. It serves as the baseline for comparison to assess the effects of selection for water efficiency on immune capabilities in broiler breeders in this study.
Comment 3, I do not understand GF-Pulp injection ?
Response 3: Thank you for your question. We believe this point has already been addressed in our response to your first comment.
Comment 4, cytokines ? if possible , Anti-inflammatory factors, such as IL-10 , shall be analyzed
Response 4: Thank you for your comment. The focus of our study was specifically on the immune capabilities of the LWCR, HWCR, and MRB lines and the production of pro-inflammatory cytokines, as these are critical for understanding the initial immune response to the LPS, PGN, and SEV. While anti-inflammatory factors like IL-10 are also important in modulating the immune response, our primary goal in this study was to evaluate the pro-inflammatory cytokine profile in response to the test materials in the different genetic lines. We appreciate your suggestion and will certainly consider analyzing anti-inflammatory cytokines in future research with these broiler lines.
Comment 5, Gram+ bacteria ? I do not find
Response 5: Thank you for your comment. To clarify, we did not administer Gram-positive bacteria but rather a component of Gram-positive bacteria, specifically peptidoglycan (PGN), which is a major cell wall component. The PGN is from Staphylococcus aureus. The primary focus of this study was to assess and compare the immune capabilities of broiler breeders selected for water efficiency. We used LPS, PGN, and SEV as immune stimulants to evaluate the birds' immune response. Independent of the immune stimulant, our goal was to examine how selection for water efficiency affects the host's immune system.
Comment 6, Male Broiler Breeders were used , why not ot use female ?
Response 6: Thank you for your question. To clarify, broiler breeder pullets (female) from the LWCR, HWCR, and MRB lines were specifically used in the third Trial, where we focused on assessing their immune response capabilities to SEV (lines 159-169).
Reviewer 2 Report
Comments and Suggestions for Authors
The authors conducted a study (3) to monitor and assess the immune profile/capabilities of the MRB, LWCR, and HWCR broiler breeder lines. Overall, selection for improved water efficiency in LWCR did not have a negative impact in the immune response capability to LPS, PGN, and killed-SEV.
Please find my recommendations below:
1. In the title please specify "Salmonella Enteritidis vaccine".
2. In the abstract, when explaining the results please specify when the results were statistically significant or non-significant according to your p-value threshold.
Please also specify this for the overarching conclusion statement of your findings in your abstract and in your conclusion section as well.
3. Line 63- What reference are you using when stating that eggs have a low cost worldwide? Egg price has increased in present day, so please re-write this sentence to better explain the message you are trying to convey and cite the source.
4. For experimental design section- Add a flow chart or table to simplify/summarize the experimental design of the three trials for your readers.
5. Line 189- Previously you referred to IgY as IgY(G) and here you just wrote IgG. Please be consistent with these details throughout the paper.
6. Line 101- When first mentioning Salmonella Enteritidis in the manuscript please refer to it properly as Salmonella enterica serovar Enteritidis(SEV).
7. Please justify in the manuscript why the authors chose a Salmonella Enteritidis serovar and not other prominent serovars for the vaccine.
Author Response
Reviewer 2
The authors conducted a study (3) to monitor and assess the immune profile/capabilities of the MRB, LWCR, and HWCR broiler breeder lines. Overall, selection for improved water efficiency in LWCR did not have a negative impact in the immune response capability to LPS, PGN, and killed-SEV.
Please find my recommendations below:
Comment 1. In the title please specify "Salmonella Enteritidis vaccine".
Response 1: We appreciate your suggestion and agree that specifying the vaccine type enhances clarity. We have revised the title, and now it mentions the Salmonella Enteritidis.
Comment 2. In the abstract, when explaining the results please specify when the results were statistically significant or non-significant according to your p-value threshold.
Please also specify this for the overarching conclusion statement of your findings in your abstract and in your conclusion section as well.
Response 2: Thank you for your suggestion. We have carefully revised the abstract and conclusion to ensure the inclusion of statistical significance when discussing specific results. However, the overall conclusion represents a synthesis of the findings rather than a single statistical comparison, so it cannot be paired with p-values. We have ensured that all statistical results supporting our conclusion are explicitly stated in the results section (e.g., p ≤ 0.05), and the rationale for their interpretation has been clearly detailed in the discussion and conclusion sections.
Comment 3. Line 63- What reference are you using when stating that eggs have a low cost worldwide? Egg price has increased in present day, so please re-write this sentence to better explain the message you are trying to convey and cite the source.
Response 3. Thank you for your comment. While egg prices have fluctuated recently due to avian influenza outbreaks, the relative cost of eggs compared to other protein-rich foods has remained stable over time. Historically, egg prices have risen during pathogen outbreaks but have returned to typical levels once the situation has been managed. To clarify this point, we have revised the sentence as you suggested and included a relevant reference to support this statement.
Comment 4. For experimental design section- Add a flow chart or table to simplify/summarize the experimental design of the three trials for your readers.
Response 4: We appreciate your suggestion. While we recognize the value of a flow chart or table, we believe that providing a detailed description of the experimental design is more suitable for this foundational research. In this research, we offer a comprehensive description of the experimental design to ensure that other researchers can fully understand and build upon it in the future. Simplifying it with a chart or table might exclude some important details that are crucial for subsequent studies. We believe the narrative approach establishes a strong foundation for future work in this area.
Comment 5. Line 189- Previously you referred to IgY as IgY(G) and here you just wrote IgG. Please be consistent with these details throughout the paper.
Response 5: Thank you for pointing this out. We intentionally introduced IgY(G) early in the manuscript to establish its classification, but our intention was always to refer to it as IgG later for consistency and readability. We have revised the manuscript to ensure consistent use of IgG.
Comment 6. Line 101- When first mentioning Salmonella Enteritidis in the manuscript please refer to it properly as Salmonella enterica serovar Enteritidis(SEV).
Response 6: Thank you for your suggestion. We have revised the manuscript to ensure that the first mention of Salmonella Enteritidis is properly written as Salmonella enterica serovar Enteritidis (SEV).
Comment 7. Please justify in the manuscript why the authors chose a Salmonella Enteritidis serovar and not other prominent serovars for the vaccine.
Response 7: Thank you for your comment. The primary focus of this study was to assess and compare the immune capabilities of broiler breeders selected for water efficiency rather than the specific characteristics of the killed Salmonella vaccine. We used a killed Salmonella vaccine as an immune stimulant to evaluate the birds' immune response. Independent of the serovar, our goal was to examine how the broiler breeder’s immune system reacts to the killed bacteria under water sustainability practices. For this reason, we did not emphasize the rationale behind selecting SEV over other serovars, as the bacteria itself was not the main focus of the study. However, we appreciate this suggestion.